

# Analysis of phase lead "anomalies" in the

# tidal response of groundwater levels

Anhua He[a,b,c], Yang Liu[d], Haixia Sun[e] Fan Zhang[d], Yanzhang Wang[a,b*], Ramesh P.

Singh[f]

[a] Key Laboratory of Geophysical Exploration Equipment Ministry of China, Jilin

University, Changchun 130026

[b] College of Instrumentation &Electrical Engineering, Jilin University, Changchun

130026

[c] National Institute of Natural Hazards, Beijing 100085

[d] Hainan Earthquake Agency, Haikou 570203

[e] Beijing Earthquake Agency, Beijing 100080

[f] School of Life and Environmental Science, Schmid College of Science and Technology,

Chapman University, Orange, CA 92866, USA

[**Abstract**]The tidal response to the groundwater level refers to an aquifer under the

influence of tidal forces, the pressure head (pore pressure) within the aquifer produces

changes that drive the alternating transportation of water between well-aquifers,

causing the rise and fall of the water level in the wells. Considering the driving

process of force and seepage of water, the groundwater level response should only

have a phase lag compared to the Earth's solid tides. However, the actual observation

data show that the phase of the groundwater level tidal response exceeded that of the

theoretical gravity tides, which is not in accordance with the commonly occurring

mechanical process of the phenomenon. Using the theory of trans-current recharge,

* Corresponding author:Yanzhang Wang. E-mail address: yanzhang@jlu.edu.cn.
First author: Anhua He. E-mail address: dqs_hah@163.com. https://orcid.org/0000-0003-1360-6602



the seepage of aquifer water was decomposed into lateral and vertical transport, and
the two kinds of "lagging" transport processes were superimposed to obtain the final
groundwater level tidal response, which may appear as an anomalous phenomenon in
which the phase is over the front after superposition. Taking the Lugu Lake well as an
example, before the Wenchuan earthquake, the phase of groundwater level was ahead
of the theoretical solid tide, indicating the existence of a transgressive aquifer,
whereas the groundwater level tidal factor declined from 0.28 mm/uGal before the
earthquake to 0.23 mm/uGal after the earthquake. The phase, from 15 min ahead in
pre-earthquake to 15 min lagged after the earthquake, combined with the theoretical
analysis it can be seen that the Wenchuan earthquake led to develop the new fissure in
the Lugu Lake well, thus permanently altering its aquifer response and changing the
permeability of the aquifer. However, the subsequent earthquakes did not produce
new fissures; only the seismic waves caused by the stress redistribution process were
observed. This co-seismic response of the groundwater level shows a step-down
phenomenon, phase analysis of the groundwater level has scientific significance for
the study of well-aquifer conditions and well-borehole seismic capacity.

[**Key Words**]Groundwater level tidal; Tide factor; The phase; co-seismic response

**1. Introduction**
The quantitative analysis of groundwater levels in unconfined boreholes as
affected by Earth solid tides and atmospheric pressure is a classical problem in the
hydrogeological and seismic subsurface fluid disciplines. Efforts have been made for



the theoretical calculations and characterization (Gulley et al., 2013; Liu et al., 2017;
Lee et al., 2017; He et al., 2020; Yan et al., 2020; Ma et al., 2023). Earth solid tides
are characterized by fixed periods and amplitudes, and the tidal response of
groundwater level can be conveniently utilized for quantitative calculation of
well-aquifer structural parameters. Cooper et al. (1965) derived an equation of motion
for the water column in the well and a method for the calculation of the aquifer
transmissivity (T) and aquifer storage rate (S). The groundwater level has zero delay
property for ground vibration in a certain frequency range (He et al., 2017), and the
groundwater level tidal response has the same frequency as the Earth's solid tides. It is
expressed as a sum of harmonics, which only differs in amplitude and phase (Cooper
et al., 1965). Hsieh et al. (1987) gave a formula for calculating the amplitude and
phase of the tidal response of the water level in wells. The amplitude of the vibration
of the water level in wells is mainly affected by the characteristic parameters of elastic
deformation of the aquifer, such as Skempton's coefficient, bulk modulus, and
Poisson's ratio (Ding et al., 2015;Arditty et al., 1978; Bredehoeft, 1967; Burbey, 2010;
Sato et al., 2006; Gao et al., 2020; Van der Kamp and Gale, 1983;All ègre et al., 2016);
and the phase is mainly related to the water flow properties of the aquifer, such as
permeability (Zhu et al., 2021; Elkhoury et al., 2006; Lai et al., 2014; Shi et al., 2014;
Xue et al., 2013; Zhu, 2021).
Assuming that the aquifer is confined, laterally extended, homogeneous, and
isotropic, Hsieh et al. (1987) utilized the diffusion equation to derive the phase lag of
the tidal response of the water level in the well as a function of aquifer permeability,
storage rate, and Earth's solids tides, and there is always a phase lag in water level
tidal response of wells because it takes a certain amount of time for the fluids in the



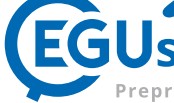

aquifer to respond to the tidal force and flow into or out of the wells. Chinese scholars
have also obtained a similar understanding through theoretical analysis, that is, the
inelastic response of the aquifer system to the tidal force leads to a phase difference in
the tidal response of the water level in the wells, the size of which mainly depends on
the radius of the borehole and the permeability of the aquifer (Zhang et al., 1991;
Zhou et al., 1993), but there is no reasonable explanation for the phase overshooting
of the tidal response of the wells' water levels. Maas and De Lange (1987) derived the
phase shift and attenuation equations in the case of a weakly permeable layer
overlying a single aquifer using the superposition principle. Studies comprehensively
show that phase lead in the tidal response of groundwater level is due to the
semi-confined aquifer with weakly permeable water quality. Studies on the conditions
for the existence of phase lead have been carried out using numerical simulations
from the theoretical aspect by above authors; however, reports on how to reflect this
in the analysis of the actual data and carry out response decomposition of the different
layers of the aquifer from the actual observation data of the water level of wells are
lacking. In the present study, our efforts are to explain the phase lead of the tidal
response of the groundwater level by combining the actual situation of the borehole
and groundwater level observation from China Earthquake Networks Center.

The water level data from the Lugu Lake seismic station (51306, Figure 1)

(N27.73 °, E100.85 °) is considered as an example for analysis. This station exposed
strata is sandstone of the upper diamictic Black Mud Whistle Formation (P2h) and the
sand and gravel layers of alluvial, flood, lacustrine, and slope deposits. Tectonically, it
is located in the two flanks of the Yanyuan Arc Tectonic Belt, and the nearby fractures
mainly include the Dog Drilling Cave - Cat's Home Village, Gaizu-Xizhu, and
Bleaching fractures. The fracture in this region is a positive fracture. The depth of the



borehole is 200.7 m, with 61 m of clay cover, under which is tuff. The depth of the
casing is 74.2 m, and process of stopping water is adopted by the piston pressure of
the casing reserved hole in the bottom section of the borehole until the cement slurry
is returned from the ground to reduce the interference of the surface water on the
water level of the borehole. The pumping experiment at the borehole completion
showed that the permeability coefficient was 0.135 m/d.

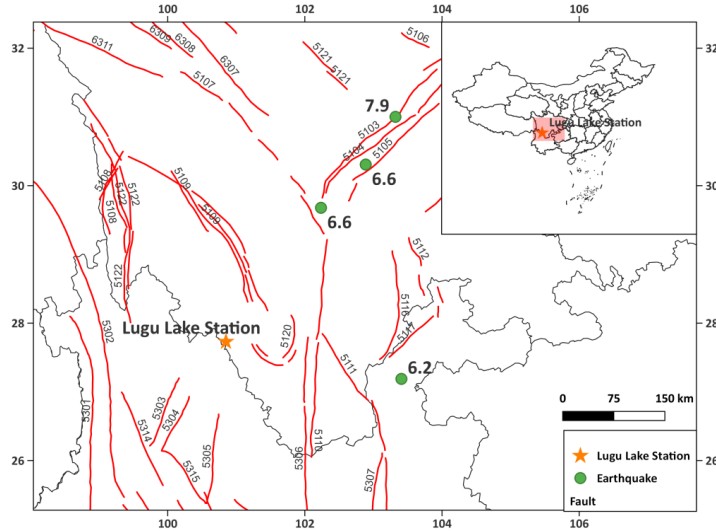


**Figure 1. Distribution of earthquakes in and around the Lugu Lake Seismic Station**

(5103, 5104, 5105: Longmenshan fault zone;    5110: Annin River fault zone; 5109: Litang-Derwu fault zone; 5122:

Jinsha River fault zone)

The water level observation of the Lugu Lake station started in December 2007.

The water level data have 1-min sampling intervals with a resolution of 1 mm. The
precise time was determined using a Global Positioning System (GPS) or Simple
Network Time Protocol (SNTP) to ensure temporal consistency with a resolution
and accuracy of 5 s. The seismic events with magnitudes greater than 6.2 (Table 1)
were selected for analysis within 500 km around the Lugu Lake station.



Table 1. Seismic events with magnitudes greater than 6.2 within 500 km around the Lugu Lake
Station

| Identifier | Time | Latitude | Longitude | Mag (mw) | Distance (km) | Location |
|---|---|---|---|---|---|---|
| EQ1 | 2008-05-12 14:28 | 31.002 | 103.322 | 7.9 | 437 | 58 km W of Tianpeng, China |
| EQ2 | 2013-04-20 08:02 | 30.308 | 102.888 | 6.6 | 350 | 56 km WSW of Linqiong, China |
| EQ3 | 2014-08-03 16:30 | 27.1891 | 103.4086 | 6.2 | 258 | 33 km WSW of Zhaotong, China |
| EQ4 | 2022-09-05 12:52 | 29.6786 | 102.236 | 6.6 | 257 | 44 km SE of Kangding, China |

**2. Method**

In order to calculate the tidal response parameters of the well water level more

accurately, a low-pass filter with a cut-off frequency of $48h^{-1}$ is used to extract the
trend of the well water level. Then, the low-frequency data is subtracted from the
original data, and only the tidal components of the well water level is retained (Figure

2).

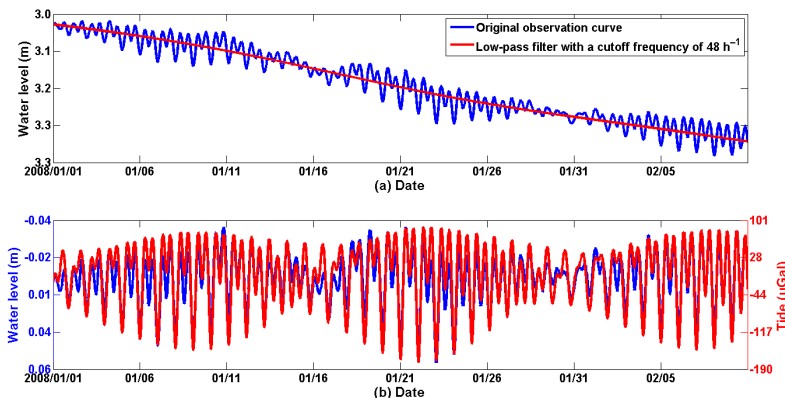


Figure 2. Tidal components of the well water level is retained

The t_tide tool provided by Pawlowicz et al. (2002) was used to calculate the

magnitude and phase of each tidal sub-wave (Figure 3), which was carried out by
comparing the groundwater level and theoretical solid gravity tide to make the





magnitude and phase comparable in the time series. For example, for the M2 wave
(period of 12.42 h), the magnitude and phase parameters of the groundwater level and
the solid gravity tide were calculated separately using the t_tide tool and then
according to Equation (1).

$$f = A_w/A_g$$

(1)

$$\varphi_d = \varphi_w - \varphi_g$$

where f is the groundwater level tidal factor of the M2 wave, $A_w$ and $A_g$ are
the groundwater level and gravity tidal oscillation amplitude in mm and uGal,
respectively. $\varphi_w$ and $\varphi_g$ are the phases of groundwater level and M2 sub-wave of
the gravity tidal wave, respectively; $\varphi_d$, the phase difference between the
groundwater level and gravity tidal wave is greater than 0 to indicate a phase lead and
less than 0 to indicate a phase lag.

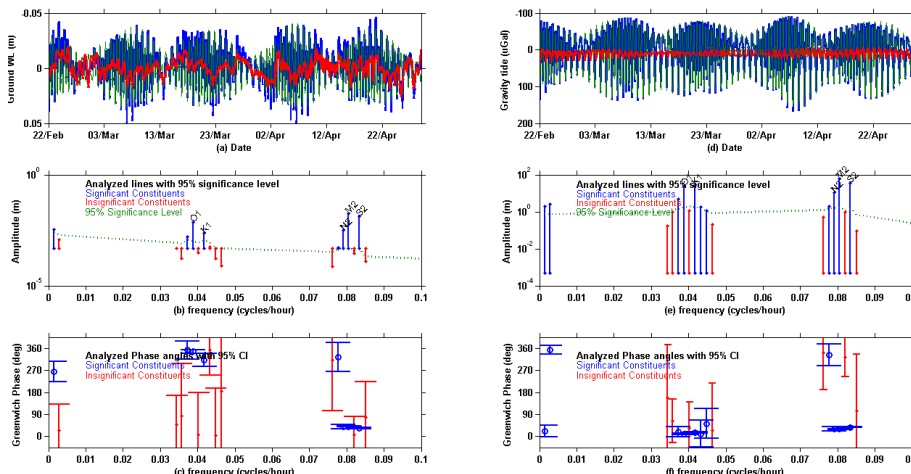

**Figure 3. Tidal parameter calculations using the t_tide tool for the Lugu Lake Station**
**groundwater level and theoretical gravity tides**



## 3. Results

The multi-year variability of water level is shown in Figure 4; the annual variation of the water level in the Lugu Lake well is approximately 1 m and a clear solid tide was recorded, that showed an annual variation pattern of high in winter and low in summer. The water level shows less disturbed, the data observed were stable and reliable, in addition to seasonal perturbations.

All four selected seismic events show significant coseismic responses, remarkably, the Wenchuan earthquake (Mw 7.9, 437 km) on May 12, 2008 caused a minor step increase in the water level of the well, the other three earthquakes caused significant step decline in the water level, and the magnitude of the step decline of the water level and of the earthquake are found to show a positive correlation.

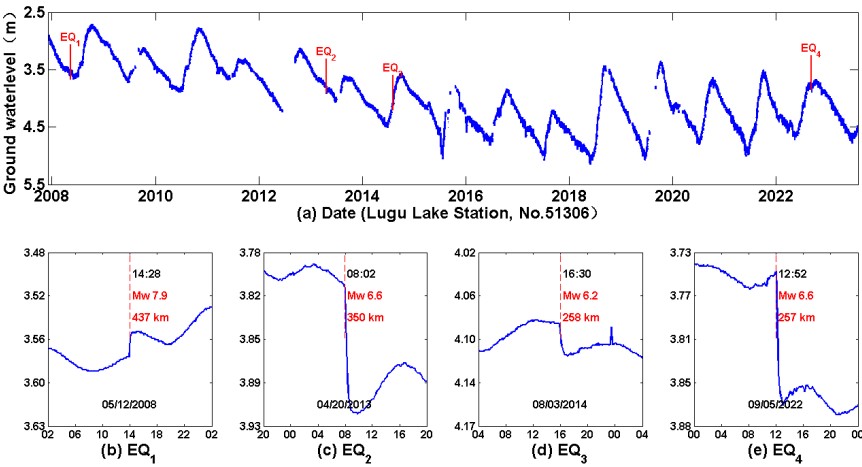

**Figure 4. Multi-year variations of water level in the Lugu Lake station and co-seismic response to earthquakes**

The used data was collected over a period of 60 days for each calculation, e.g.,



the result for the data collected from February 21–April 20, 2008 is showed in Table

2.

Table 2. The results for the data collected from February 21–April 20, 2008

| tide | Period (h) | Groundwater level demodulation | | | | Gravitational tidal demodulation | | | |
|------|------------|------|---------|------|---------|------|---------|------|---------|
|      |            | amp | amp_err | pha | pha_err | amp | amp_err | pha | pha_err |
| MM   | 661.29 | 3.3  | 3 | 264.7 | 41.2  | 1.91  | 0.80 | 20.8  | 25.8  |
| MSF  | 354.37 | 1.2  | 2 | 23.2  | 104.8 | 2.63  | 0.83 | 354.2 | 15.2  |
| ALP1 | 29.07  | 0.5  | 1 | 48.1  | 101.5 | 0.18  | 1.15 | 157.9 | 213.3 |
| 2Q1  | 28.01  | 0.2  | 1 | 84.5  | 200.2 | 0.99  | 1.51 | 62.8  | 105.3 |
| *Q1  | 26.87  | 1.6  | 1 | 352.4 | 35.4  | 4.77  | 1.52 | 19.6  | 17.6  |
| *O1  | 25.82  | 7.2  | 1 | 346.3 | 8.5   | 25.16 | 1.56 | 9.4   | 3.3   |
| NO1  | 24.83  | 0.3  | 1 | 6.4   | 166.6 | 1.20  | 1.71 | 32.5  | 99.5  |
| *K1  | 23.93  | 2.4  | 1 | 311.6 | 28.3  | 27.94 | 1.68 | 14.8  | 3.5   |
| J1   | 23.1   | 0.6  | 1 | 352.7 | 105.7 | 1.73  | 1.50 | 10.5  | 57.9  |
| OO1  | 22.31  | 0.2  | 1 | 4.0   | 182.9 | 1.10  | 1.12 | 52.6  | 64.6  |
| UPS1 | 21.58  | 0.1  | 0 | 185.6 | 205.8 | 0.20  | 0.74 | 24.7  | 193.6 |
| EPS2 | 13.13  | 0.1  | 0 | 312.0 | 208.3 | 0.52  | 1.22 | 341.5 | 160.0 |
| MU2  | 12.87  | 0.5  | 0 | 324.5 | 52.5  | 2.00  | 1.57 | 332.0 | 42.3  |
| *N2  | 12.66  | 3.3  | 1 | 40.2  | 9.1   | 11.16 | 1.85 | 30.5  | 9.0   |
| **\*M2** | **12.42** | **17.4** | **1** | **39.7** | **1.7** | **59.10** | **1.73** | **30.5** | **1.8** |
| L2   | 12.19  | 0.3  | 0 | 5.9   | 75.4  | 0.92  | 1.07 | 323.9 | 78.9  |
| *S2  | 12     | 13.3 | 1 | 33.6  | 2.0   | 34.89 | 1.62 | 37.8  | 2.5   |
| ETA2 | 11.75  | 0.1  | 0 | 78.9  | 133.8 | 0.09  | 0.74 | 104.4 | 221.8 |
| MO3  | 8.39   | 0.1  | 0 | 19.9  | 168.4 | 0.21  | 0.08 | 71.4  | 20.6  |
| M3   | 8.28   | 0.2  | 0 | 88.3  | 67.5  | 1.00  | 0.09 | 47.8  | 4.8   |
| MK3  | 8.18   | 0.2  | 0 | 112.6 | 69.6  | 0.03  | 0.06 | 186.6 | 116.1 |
| *SK3 | 7.99   | 0.3  | 0 | 236.8 | 35.5  | 0.04  | 0.06 | 246.9 | 94.7  |
| MN4  | 6.27   | 0.1  | 0 | 36.5  | 54.8  | 0.02  | 0.02 | 112.6 | 41.8  |
| M4   | 6.21   | 0.1  | 0 | 46.4  | 100.3 | 0.02  | 0.01 | 72.7  | 37.6  |
| SN4  | 6.16   | 0.1  | 0 | 1.2   | 74.8  | 0.03  | 0.02 | 7.3   | 28.3  |
| MS4  | 6.1    | 0.1  | 0 | 86.7  | 65.9  | 0.02  | 0.02 | 236.1 | 44.8  |
| S4   | 6      | 0    | 0 | 246.9 | 119.7 | 0.03  | 0.01 | 118.4 | 28.9  |
| 2MK5 | 4.93   | 0    | 0 | 74.3  | 157.2 | 0.02  | 0.01 | 176.3 | 16.8  |
| 2SK5 | 4.8    | 0    | 0 | 35.2  | 156.1 | 0.02  | 0.01 | 220.7 | 20.0  |
| 2MN6 | 4.17   | 0    | 0 | 258.9 | 122.3 | 0.02  | 0.01 | 302.0 | 25.9  |
| M6   | 4.14   | 0    | 0 | 206.7 | 161.9 | 0.01  | 0.01 | 174.6 | 31.6  |
| 2MS6 | 4.09   | 0.1  | 0 | 66.3  | 45.5  | 0.02  | 0.01 | 54.5  | 23.6  |
| 2SM6 | 4.05   | 0    | 0 | 275.0 | 89.4  | 0.01  | 0.01 | 232.5 | 39.7  |
| 3MK7 | 3.53   | 0    | 0 | 265.0 | 128.0 | 0.01  | 0.00 | 315.2 | 11.1  |
| M8   | 3.11   | 0.1  | 0 | 36.9  | 48.7  | 0.01  | 0.00 | 357.6 | 3.4   |

Calculating the M2 wave tidal parameters for February 21-April 20, 2008,
according to Equation (1):





$$f = \frac{A_w}{A_g} = \frac{17.4mm}{59.10uGal} = 0.2944 \; mm/uGal$$

$$\varphi_d = \varphi_w - \varphi_g = 39.7 - 30.5 = 9.2° = \frac{9.2}{360} * 12.42 * 60 \approx 19 \; min$$

Following this procedure, the M2 fractional wave eigenvalues of the
groundwater level for each of the four seismic events (Table 1) were calculated and
the results are shown in Figure 5.

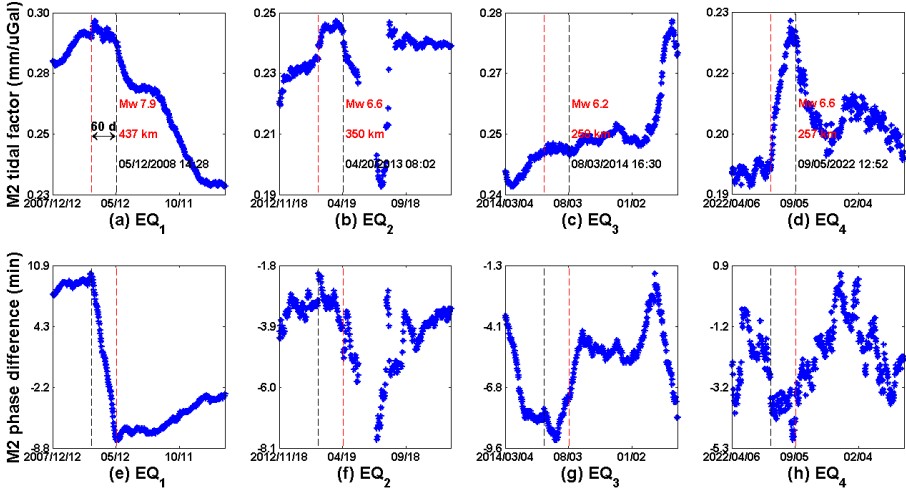

**Figure 5. M2 fractional wave eigenvalues in four seismic events at the Lugu Lake Station**
**well water level**
The Wenchuan earthquake (05/12/2008, Mw 7.9) shows the greatest impact on
the eigenvalues of the M2 sub-wave at Lugu Lake station. Its M2 wave tidal factor
decreased from 0.28 mm/uGal prior to the earthquake to 0.23 mm/uGal, especially for
the relative phase, which changed from an average phase lead of 10 min prior to the
earthquake to an average phase lag of 9 min after the earthquake. In addition, the M2
sub-wave tidal factor size and phase lead size show an increasing trend prior to the





Wenchuan earthquake. It was difficult to affirm if the trend was anomalous prior to
the Wenchuan earthquake due to the limited pre-earthquake observation data. The
tidal factor size and relative phase changes were less evident in the other three
earthquakes.
**4. Discussion**

The moon, the sun, and other celestial bodies exert gravitational force on Earth's

mass points, which causes the volume of the Earth to change. The volume of the solid
skeleton of the aquifer in the Earth's crust changes accordingly, resulting in changes in
the aquifer's pressure head (pore pressure) and the consequent rise and fall of the
water level in the well. For the ideal horizontal stratified pressurized aquifer model,
the partial differential equation for the effect of solid tidal body strain on the pressure
head of the pressurized aquifer can be derived from the theory of elasticity and theory
of groundwater dynamics as follows:

$$\frac{\partial H}{\partial t} = \frac{K}{S_s} \nabla^2 H - \frac{1}{S_s} \frac{\partial \Theta}{\partial t} \qquad (2)$$

In the above equation, $H$ is the pressure head within the aquifer, $t$ is time, $K$

is the permeability coefficient of the aquifer, $S_s$ is the unit water storage coefficient
of the aquifer, and $\Theta$ is the solid tidal body strain. Under the condition of no drainage,
the simplified solution of equation (2) is:

$$H_n = \frac{-1}{S_s} \sum_i A_{ti} \cos(\omega_{ti} t - \varphi_{ti}) + H_0 \qquad (3)$$

In the above equation, $A_{ti}$, $\omega_{ti}$, $\varphi_{ti}$ are the amplitude, angular frequency, and

initial phase of the ith harmonic of the solid tidal body strain at the moment $t$,



respectively, and $H_0$ is the average pressure head of the aquifer. When there is a
seepage influence between well-aquifer and only the case of solid tidal, an angular
frequency$(\omega)$ harmonic is considered, and the equation solution can be changed to:

$$H_w = H_n + H_0\cos(\omega t + \varphi_w + \Psi) \qquad (4)$$

Where $H_w$ is the pressure head of the well water column in the pressurized
aquifer (groundwater level), $\varphi_w$ is the phase angle of the groundwater level to the
solid tide response. It can be seen that a certain independent sub-tidal wave in the
groundwater level can be simplified to a sine-cosine function for calculation.
Regarding a well-aquifer model, the water barrier (impermeable layer) covering
the top of the original aquifer is changed into a leakage aquifer, i.e., a semi-confined
aquifer, which is a soil layer with poor permeability, and a head difference exists
between the aquifer and adjacent aquifer. Although the permeability of the soil layer
of the leaky aquifer is poor, under the condition that a head difference exists between
the aquifer and neighboring aquifer, water seepage between the aquifer and leaky
aquifer also occurs. Owing to the wide area of distribution, the total amount of water
is considerable. Compared with the junction area between the aquifer, interface
between the transgressive aquifer, and the borehole is negligible, i.e., the amount of
water alternation between the transgressive aquifer and borehole is negligible.



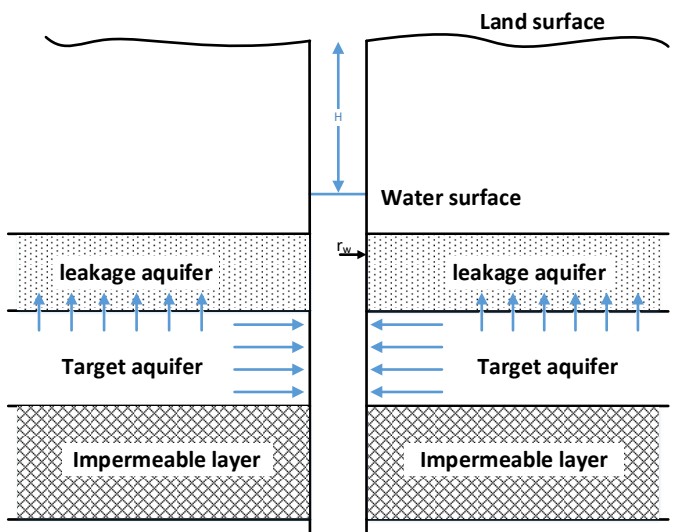


**Figure 6. Well-Aquifer Modeling**


A completely confined aquifer (Figure 6), without considering the friction and
other factors, the water will directly squeeze into the borehole and its pressure will be
transferred to the borehole without any loss, that raise the water level of the well. In a
transgressive aquifer, the pressure within will be shared partially, affecting the
efficiency of tidal response of the water level in the borehole. The efficiency of the
tidal response of the groundwater level is affected by the presence of a transgressive
aquifer. Compared to the simple one-dimensional well-aquifer structure, the water
level in a borehole with a transgressive aquifer can be decomposed into two simple
harmonic oscillations in the same direction and frequency.

$$x_1 = A_1 \cos(\omega t + \varphi_1)$$

(5)

$$x_2 = A_2 \cos(\omega t + \varphi_2)$$

Where $x_1$ is the recharge of water from the ideal aquifer to the borehole and $x_2$





is the loss of water from the aquifer to borehole owing to leakage to the
semi-pressurized aquifer; the magnitude of the rise in the water level in the borehole
can be expressed as $x = x_1 - x_2$:

$$x = x_1 - x_2$$

$$= A_1 \cos(\omega t + \varphi_1) - A_2 \cos(\omega t + \varphi_2)$$

$$= (A_1 \cos\varphi_1 \cos\omega t - A_1 \sin\varphi_1 \sin\omega t) - (A_2 \cos\varphi_2 \cos\omega t - A_2 \sin\varphi_2 \sin\omega t)$$

$$= (A_1 \cos\varphi_1 - A_2 \cos\varphi_2)\cos\omega t - (A_1 \sin\varphi_1 - A_2 \sin\varphi_2)\sin\omega t$$

taking $A\cos\varphi = A_1 \cos\varphi_1 - A_2 \cos\varphi_2$ and $A\sin\varphi = A_1 \sin\varphi_1 - A_2 \sin\varphi_2$
Then:

$$x = A\cos\varphi\cos\omega t - A\sin\varphi\sin\omega t = A\cos(\omega t + \varphi) \qquad (6)$$

From Equation 6, it can be seen that the frequency of the synthesized simple
harmonic continues in the same direction and its frequency remains unchanged. The
amplitude and phase of the combined vibration are:

$$A = \sqrt{(A\sin\varphi)^2 + (A\cos\varphi)^2}$$

$$= \sqrt{(A_1 \sin\varphi_1 - A_2 \sin\varphi_2)^2 + (A_1 \cos\varphi_1 - A_2 \cos\varphi_2)^2} \qquad (7)$$

$$= \sqrt{A_1^2 + A_2^2 - 2A_1 A_2 \cos(\varphi_1 - \varphi_2)}$$

$$\tan\varphi = \frac{A\sin\varphi}{A\cos\varphi} = \frac{A_1 \sin\varphi_1 - A_2 \sin\varphi_2}{A_1 \cos\varphi_1 - A_2 \cos\varphi_2} \qquad (8)$$

Equation 8 shows that the phase of the combined vibration is related to the
respective phases of the sub-vibrations and individual amplitude of the sub-vibration.
Here, we considered $M_2$ wave as an example, the simulations, procedure and





performance are shown in Table 3.
Table 3. Calculations for simulating the amplitude and phase of water level oscillations in wells
with transgressive aquifers

| Steps | Content | Illustrate |
|---|---|---|
| 1 | $td = 1 \times cos(2\pi f\tau)$ | Assuming the M2 sub-wave Earth solid tide, in which $f = 1.9324 cpd$ (cpd - cycle per day) |
| 2 | $wl\_ideal = 1 \times cos(2\pi f\tau - 0.1\pi)$ | Groundwater level vibration under an ideal pressurized aquifer structure, which is assumed to be in phase with the solid tide with a lag of $0.1\pi$, indicating a lag of about 37 minutes |
| 3 | $wl\_leakage\_i = 1 \times cos(2\pi f\tau - 0.1 \times i \times \pi)$ | Assuming that the transgressive aquifer causes the groundwater level to vibrate with an amplitude of 1, except that the phase is $0.1 \times i \times \pi$ $(i = 1,2,3,...)$ |
| 4 | $wl\_leakage\_j = 1 * j *$ $\times cos(2\pi f\tau - 0.2 \times \pi)$ | Assuming that the phase of the transgressive aquifer is fixed at a lag of $0.2\pi$ s, the amplitude takes the value of $1 * j$ $(j = 1,0.9,0.8...)$ |
| 5 | $wl = wl\_ideal - wl\_leakage\_i$ | Synthetic groundwater level |

Considering the pressure of the target aquifer drives the recharge of the overflow
aquifer, its phase will only be equal to or lag behind that of the target aquifer, and
under the premise of keeping the amplitude of the overflow aquifer unchanged,
$i = 1,2,3...$ were considered. The corresponding values of the phases are
$-0.1\pi$ ,$-0.2\pi$, $-0.3\pi$ ..., and the results of the calculations are as follows:





| $i$ | Range | Phase | $j$ | Range | Phase |
|---|---|---|---|---|---|
| 1 | 0 | —— | 1 | 0.313 | -1.100 |
| 2 | 0.313 | -1.100 | 0.9 | 0.313 | -0.779 |
| 3 | 0.618 | -0.942 | 0.8 | 0.344 | -0.488 |
| 4 | 0.908 | -0.785 | 0.7 | 0.398 | -0.260 |
| 5 | 1.176 | -0.628 | 0.6 | 0.468 | -0.093 |
| 6 | 1.414 | -0.471 | 0.5 | 0.547 | 0.028 |

236  Figure 7 shows $i$=1 indicates that the target aquifer pressure is all converted to

237 the leaky aquifer; thus, the water level in the borehole will remain unchanged, which

238 is only the ideal state. With the increase of the value of $i$, the phase lag in the leaky

239 aquifer gets progressively larger, and the synthetic phase gets progressively closer to

240 the theoretical value of the gravitational tidal phase. However, the phase is always in a

241 "lead" state. Simultaneously, as the value of $i$ increases, the synthesized amplitude

242 increases in magnitude, and the magnitude may also exceeds the magnitude of the

243 ideal pressurized aquifer structure.

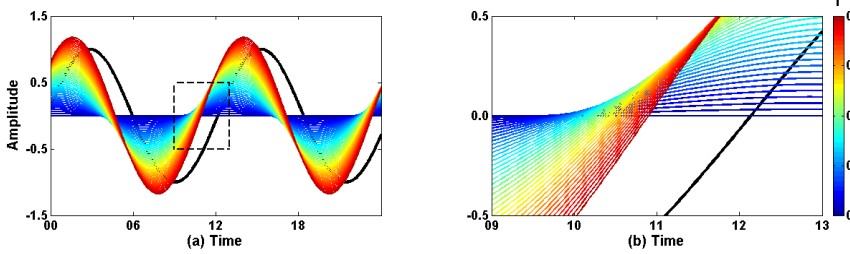

245 **Figure 7. Water level changes in synthetic wells when the leaky aquifer amplitude is**

246 **held constant and phase lag shows different values**

247  Next, the phase value of the leaky aquifer is fixed (here, it is fixed to $-0.2\pi$,





lagging behind the theoretical value of gravity tides by $0.2\pi$). Its amplitude is
changed by $j = 1$,$0.9$,$0.8$ ..., from the calculation results (Figure 8), the synthetic
vibration phase lead becomes progressively smaller as the value of j decreases.
Nevertheless, the synthetic vibration phases lead the theoretical solid-tide phases
under the prerequisite of j>0.5, and lagging occurs only when the phase of synthetic
vibration is j＜0.5; the synthetic vibration phase lags. Simultaneously, as the value of
j decreases, the amplitude of the synthetic vibration progressively increases.

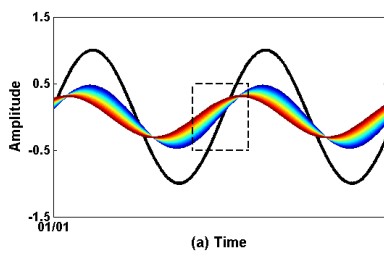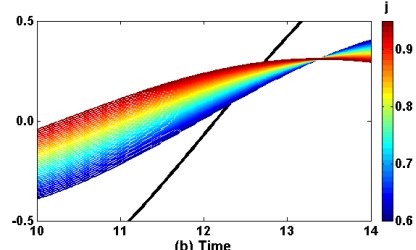

**Figure 8. Water level changes in synthetic borehole when the phase lag of the leaky**
**aquifer is held constant and the magnitude varies**

The results shown in Figure 7&8 clearly show that closer the phase and

amplitude of the leaky aquifer are to those of the ideal aquifer, the more the phase of
its synthetic vibration lead to the theoretical solid gravity tide.
**5.  Conclusions**

Based on the theoretical calculation our results show the large phase lag of the

leaky aquifer, the closer the synthetic phase to the theoretical value of gravity tidal
phase. But the phase always remains "phase lead"; simultaneously, the synthetic
amplitude increased progressively, and the amplitude may even exceed that of the
ideal pressurized aquifer structure. In addition, the smaller the oscillation amplitude of



the leaky aquifer, the smaller the synthetic vibration phase lead. However, under the
premise of j>0.5 (j is the amplitude ratio of the leaky aquifer relative to the ideal
aquifer), the synthetic vibration phase lead was in excess to the theoretical solid tidal
phase. The synthetic vibration phase lag occurred only when j<=0.5. Meanwhile, the
synthetic vibration amplitude became progressively larger as the value of j decreased.

The Wenchuan earthquake caused the M2-wave tidal factor of the water level in

the Lugu Lake Station well to decrease from 0.28 mm/uGal prior to the earthquake to
0.23 mm/uGal; the phase lead from an average of 10 min of pre-seismic overshooting
to an average of 9 min of post-seismic lagging. The pre-earthquake phase leads over
the theoretical gravity tide, indicates the existence of a leaky aquifer in the Lugu Lake
well. After the earthquake, the phase changed from lead to lag, which can only be due
to either the disappearance of the leaky aquifer or lagging of the synthetic vibrational
phase because the amplitude ratio of the leaky aquifer relative to the ideal aquifer is
<=0.5. The leaky aquifer, as an actual aquifer, cannot disappear because of the
earthquake. However, the earthquake can only change the amplitude ratio of the leaky
aquifer relative to the ideal aquifer.

The Wenchuan earthquake caused an increase in the permeability of the aquifer

in the water wells of Lugu Lake station, and this increase may be due to the
development of new fractures in the aquifer (Tokunaga, 1999; Zhang et al., 2019; Sun
et al., 2015) that enhances the permeability owing to the scouring of the fracture water
in the aquifer caused by the seismic wave (Yoshimi and Oh-Oka, 1975; Dobry et al.,
1982; Vucetic, 1994; Hsu and Vucetic, 2004; Lai et al., 2004; Wang and Manga, 2010).
Owing to the new fractures, the former shows a permanent increase in permeability.
Simultaneously, the latter will accumulate over time and revert back to the original





level in the aquifer because of the accumulation of groundwater minerals. From the
effect of several earthquakes, except for the Wenchuan earthquake, which caused the
groundwater level to step up, the subsequent earthquakes caused the groundwater
level to step down. However, the Wenchuan earthquake changed the permeability of
the Lugu Lake well aquifer, which did not return to the pre-earthquake level, in line
with the permanent increase in permeability and development of new fissures.
Furthermore, the other three earthquakes did not show a permanent change in
permeability and development of new fissures.

The Wenchuan earthquake led to the generation of new fissures in the Lugu Lake,

which led to the conduction of deep high-pressure water into the aquifer, resulting in a
co-seismic response of step-up; the subsequent earthquakes did not generate new
fissures but seismic waves only caused fissure water flushing that enhanced the
permeability of the aquifer and a step-down in the water level of the borehole when
boreholes were close to the discharge zone (He and Singh, 2018). The M2 fractional
wave tidal factor size and phase lead size prior to the Wenchuan earthquake showing
an increasing trend; however, the anomalous nature of this trend prior to the
Wenchuan earthquake can not be affirmed due to the limited pre-earthquake
observation data.
**Declaration of Competing Interest**

The authors declare that they have no known competing financial interests or

personal relationships that could have appeared to influence the work reported in this
paper.
**Code/Data availability**



At present, data and code can only be submitted to reviewers in the form of
attachments.
**Author contribution**
Anhua He carry out the design of the manuscript ideas, Yang Liu & Fan Zhang
completed data collection and editing work, Haixia Sun completed partial code,
Yanzhang Wang and Ramesh P. Singh review and improve the manuscript.
**Acknowledgments**
The present study was supported financially by National Natural Science
Foundation of China (41772256) and National Key R&D Program of China,
(2018YFC1503803). The authors thank the China Earthquake Network Center for
providing the water-level data and the United States Geological Survey for providing
the seismic data.
[**References**]
Allègre, V., Brodsky, E. E., Xue, L., Nale, S. M., Parker, B. L., & Cherry, J. A. (2016).

Using earth‐tide induced water pressure changes to measure in situ

permeability: A comparison with long‐term pumping tests. Water Resources

Research, 52(4), 3113-3126.

Arditty, P. C., Ramey Jr, H. J., & Nur, A. M. (1978, October). Response of a closed

well-reservoir system to stress induced by earth tides. In SPE Annual Technical

Conference and Exhibition? (pp. SPE-7484). SPE.

Bredehoeft, J. D. (1967). Response of well‐aquifer systems to Earth tides. Journal of



Geophysical Research, 72(12), 3075-3087.
Burbey, T. J. (2010). Fracture characterization using Earth tide analysis. Journal of

Hydrology, 380(3-4), 237-246.

Cooper Jr, H. H., Bredehoeft, J. D., Papadopulos, I. S., & Bennett, R. R. (1965). The

response of well‐aquifer systems to seismic waves. Journal of Geophysical

Research, 70(16), 3915-3926.

Ding, F. H., Han, X. L., Ha, Y. Y., & Dai Yong, L. Y. (2015). Relationship of porosity

and volume compression coefficient of solid skeleton and water in artesian well

aquifer. Earth Science: Journal of China University of Geosciences, 40(7),

1248-1253.

Dobry, R., Ladd, R. S., Yokel, F. Y., Chung, R. M., & Powell, D. (1982). Prediction of

pore water pressure buildup and liquefaction of sands during earthquakes by the

cyclic strain method (Vol. 138, p. 150). Gaithersburg, MD: National Bureau of

Standards.

Elkhoury, J. E., Brodsky, E. E., & Agnew, D. C. (2006). Seismic waves increase

permeability. Nature, 441(7097), 1135-1138.

Gao, X., Sato, K., & Horne, R. N. (2020). General solution for tidal behavior in

confined and semiconfined aquifers considering skin and wellbore storage effects.

Water Resources Research, 56(6), e2020WR027195.

Gulley, A. K., Ward, N. F. D., Cox, S. C., & Kaipio, J. P. (2013). Groundwater

responses to the recent Canterbury earthquakes: a comparison. Journal of



Hydrology, 504, 171-181.
He, A., Deng, W., Singh, R. P., & Lyu, F. (2020). Characteristics of hydroseismograms
in Jingle well, China. Journal of Hydrology, 582, 124529.
He, A., Fan, X., Zhao, G., Liu, Y., Singh, R. P., & Hu, Y. (2017). Co-seismic response
of water level in the Jingle well (China) associated with the Gorkha Nepal (Mw
7.8) earthquake. Tectonophysics, 714, 82-89.
He, A., & Singh, R. P. (2019). Groundwater level response to the Wenchuan
earthquake of May 2008. Geomatics, Natural Hazards and Risk, 10(1), 336-352.
He, A., Singh, R. P., Sun, Z., Ye, Q., & Zhao, G. (2016). Comparison of regression
methods to compute atmospheric pressure and earth tidal coefficients in water
level associated with Wenchuan Earthquake of 12 May 2008. Pure and Applied
Geophysics, 173, 2277-2294.
Hsieh, P. A., Bredehoeft, J. D., & Farr, J. M. (1987). Determination of aquifer
transmissivity from Earth tide analysis. Water resources research, 23(10),

1824-1832.

Hsu, C. C., & Vucetic, M. (2004). Volumetric threshold shear strain for cyclic
settlement. Journal of geotechnical and geoenvironmental engineering, 130(1),

58-70.

Lai, G., Ge, H., Xue, L., Brodsky, E. E., Huang, F., & Wang, W. (2014). Tidal
response variation and recovery following the Wenchuan earthquake from water
level data of multiple wells in the nearfield. Tectonophysics, 619, 115-122.





Lee, S. H., Cheong, J. Y., Park, Y. S., Ha, K., Kim, Y., Kim, S. W., & Hamm, S. Y.
(2017). Groundwater level changes on Jeju Island associated with the Kumamoto
and Gyeongju earthquakes. Geomatics, Natural Hazards and Risk, 8(2),

1783-1791.

Liu CP, Liao X, Shi Y, Tang YD. 2017. Crustal stress and groundwater dynamic
response (in Chinese). Beijing: Seismological Press.
Ma, Y., Wang, G., Shi, Z., Yan, R., & Yu, H. (2023). Groundwater level and
temperature changes following the great Tangshan earthquake of 1976 near the
epicenter. Geomatics, Natural Hazards and Risk, 14(1), 2197103.
Maas, C., & De Lange, W. J. (1987). On the negative phase shift of groundwater tides
near shallow tidal rivers—The Gouderak anomaly. Journal of Hydrology, 92(3-4),

333-349.

Pawlowicz, R., Beardsley, B., & Lentz, S. (2002). Classical tidal harmonic analysis
including error estimates in MATLAB using T_TIDE. Computers & Geosciences,

28(8), 929-937.

Rasmussen, T. C., & Crawford, L. A. (1997). Identifying and removing barometric
pressure effects in confined and unconfined aquifers. Groundwater, 35(3),

502-511.

Sato, M., Fujita, S., Saito, A., Ikeda, Y., Kitazawa, H., Takahashi, M., ... & Aizawa, Y.
(2006). Increased incidence of transient left ventricular apical ballooning
(so-calledTakotsubo'cardiomyopathy) after the mid-Niigata Prefecture
earthquake. Circulation journal, 70(8), 947-953.



Shi, Z., Wang, G., Wang, C. Y., Manga, M., & Liu, C. (2014). Comparison of
hydrological responses to the Wenchuan and Lushan earthquakes. Earth and
Planetary Science Letters, 391, 193-200.

Sun, X., Wang, G., & Yang, X. (2015). Coseismic response of water level in
Changping well, China, to the Mw 9.0 Tohoku earthquake. Journal of Hydrology,
531, 1028-1039.

Tokunaga, T. (1999). Modeling of earthquake-induced hydrological changes and
possible permeability enhancement due to the 17 January 1995 Kobe Earthquake,
Japan. Journal of Hydrology, 223(3-4), 221-229.

Toll, N. J., & Rasmussen, T. C. (2007). Removal of barometric pressure effects and
earth tides from observed water levels. Groundwater, 45(1), 101-105.

Van der Kamp, G., & Gale, J. E. (1983). Theory of earth tide and barometric effects in
porous formations with compressible grains. Water Resources Research, 19(2),
538-544.

Vucetic, M. (1994). Cyclic threshold shear strains in soils. Journal of Geotechnical
engineering, 120(12), 2208-2228.

Wang, C. Y., Doan, M. L., Xue, L., & Barbour, A. J. (2018). Tidal response of
groundwater in a leaky aquifer—Application to Oklahoma. Water Resources
Research, 54(10), 8019-8033.

Wang C. Y., Manga M. 2010. Earthquakes and water. Lecture Notes in Earth Sciences
114, Berlin: Springer-Verlag.





Wood, M. D., Allen, R. V., & Allen, S. S. (1973). Methods for prediction and
evaluation of tidal tilt data from borehole and observatory sites near active faults.
Philosophical Transactions for the Royal Society of London. Series A,
Mathematical and Physical Sciences, 245-252.
Xue, L., Li, H. B., Brodsky, E. E., Xu, Z. Q., Kano, Y., Wang, H., ... & Huang, Y.
(2013). Continuous permeability measurements record healing inside the
Wenchuan earthquake fault zone. Science, 340(6140), 1555-1559.
Yan, R., Wang, G., Ma, Y., Shi, Z., & Song, J. (2020). Local groundwater and tidal
changes induced by large earthquakes in the Taiyuan Basin, North China from
well monitoring. Journal of Hydrology, 582, 124479.
Yan, X., Shi, Z., Wang, G., Zhang, H., & Bi, E. (2021). Detection of possible
hydrological precursor anomalies using long short-term memory: A case study of
the 1996 Lijiang earthquake. Journal of Hydrology, 599, 126369.
Yoshimi, Y., & Oh-oka, H. (1975). Influence of degree of shear stress reversal on the
liquefaction potential of saturated sand. Soils and foundations, 15(3), 27-40.
Zhang, S., Shi, Z., & Wang, G. (2019). Comparison of aquifer parameters inferred
from water level changes induced by slug test, earth tide and earthquake–A case
study in the three Gorges area. Journal of Hydrology, 579, 124169.
Zhang, Z. D., Wang, C. W., & Gao, Y. B. (1991). The relationship between variation
of radius in the well Shandong Province No. 7 and response to Earth tide.
Chinese Journal of Geophysics, 34(2), 203-209.
Zhou, K. G., Li, S. L., Tan, S. L., Li, H., Rong, J. D., et al. (1993). On phase shift in



observation of tidal fluctuation in a deep well. Crustal Deformation and
Earthquake, 13(3), 18-24.
Zhu, A. Y. (2021). Unsaturated Flow Influences the Response of Leaky Aquifer to
Earth Tides. Lithosphere, 2021(Special 3), 6415482.
Zhu, J. B., Kang, J. Q., Elsworth, D., Xie, H. P., Ju, Y., & Zhao, J. (2021). Controlling
induced earthquake magnitude by cycled fluid injection. Geophysical Research
Letters, 48(19), e2021GL092885.