# Peer review of "tidal response of groundwater levels"

_EGUsphere, 2023_

## Author Comment (AC3)

This manuscript documented a phase lead phenomenon of groundwater level in response to earth tide and try to explain what the phase advance occur.

To be honest, I do not think the phase advance is new and unexplainable, and there are several published papers have dealt the issue (pelease refer to Barbour et al., 2019; Gao et al., 2020; Valois et al. 2023 .Wang et al. 2018 ....). In these papers, they discussed the changes of Taqu, Tcon, Saqu, Scon would have signficant impact on the phase shift. Such findings provide a basic physical mechanism of the phase shift. The discussion of your combination of two signals (x1, x2) only provides an exterior discussion on the changes of tidal wave, and ignored the physical nature of such phase advance phenomenon. Furthermore, your synthetic wave could not fit the theories M2 wave in Fig.7 and Fig.8, which means the fixed phase shift of leaky aquifer or fixed amplitude may not able to explain your observations.

Answer: We would like to thank the anonymous reviewer for their insightful comments. The quality of the manuscript was significantly improved based on these suggestions. Due to some wording and translation errors in the original manuscript, a considerable amount of the information was misunderstood by readers. The revised manuscript has been carefully checked, and a professional English translation company was invited to revise the manuscript. We acknowledge that many people have carried out similar studies on phase advancement. However, this study mainly focused on using actual observation data to demonstrate that the phase change caused by earthquakes is due to new fractures. The effect of new fractures is similar to that of leaky aquifers. This study explores this issue in another direction by combining mathematical methods with actual observation data, and the calculation method can fit the actual observation process. In the original manuscript, to clarify the display, the phase lag of the tidal response of the aquifer was set to -0.1 π, which is larger than the actual observation. After setting the value smaller, we can fully achieve the actual phase leading effect, as shown in the new figures 7 & 8.

Second, the tidal phase shift results showed in this manuscript is opposite with other published studies (Lai et al., 2014; Lai et al., 2016) , the phase shift of Lugu Lake(LGH in their paper) is around -10° before the Wenchuan earthquake, and increased to 10° after the earthquake. And it is unclear how you set the parameters in the tidal analysis, I am also not quite sure about the result from T_tide. The widely used tidal analysis programs are Baytap-08, HALS method etc, I think you should rechecked your setting and the results.

Answer: We have carefully considered this question. Through our comparison of several methods (T_tide, Baytap-08, HALS method, etc.), the results obtained by different calculation methods are almost identical except for minor differences. According to the statement "Fourier analysis provides a robust means of estimating equilibrium conclusion status" (Turnadge et al., 2019), Fourier analysis has been directly used to calculate the tidal response parameters.

Turnadge, C., Crosbie, R. S., Barron, O., and Rau, G. C. (2019). Comparing Methods of Barometric Efficiency Characterization for Specific Storage Estimation, Groundwater, 57, 844–859, https://doi.org/10.1111/gwat.12923.

Minor comments:

1. I cannot understand many terminologies in your manuscript and I think you should pay much more attention about them. i.e. trans-current recharge, transgressive aquifer, semi-[pressurized aquifer and so on.....

   Answer: This is a very good suggestion. First, we standardized these concepts and used professional vocabulary; second, we have provided a unified interpretation of these professional terms in the methodology. Please see the third paragraph of the Introduction.

2. I would strongly ask you to seek help from a native speaker to help you improve the language of this manuscript.

   Answer: Thank you for this advice. We have invited a professional translation company to improve the manuscript's language, and your advice has been conveyed to the relevant staff.

3. A borehole lithology is required to show the aquifer type of this well, where does the semi-confined layer exists? And also the hydrogeological setting of the study area should be presented.

   Answer: Based on the original drilling data and information provided by Lai et al. (2016), we supplemented the borehole lithology and hydrogeological setting data for the study area. Please see Figure 2 and the associated content.

4. Line 43-44, borehole has no unconfined or confined characteristics, here should be confined aquifer, and not unconfined aquifer, and it is very difficult for the unconfined aquifer to record tidal signal.

   Answer: We apologize for any confusion about this issue, which is a language translation error. The original meaning is that the wellhead is unconfined and open. We did not mean to indicate that it is an "unconfined aquifer." We have corrected this error in the revised manuscript.

5. Line 53-54 groundwater level could record the tidal signal, but it also contains many other frequency components of other signals.

Answer: I strongly agree with this comment. The groundwater level contains many other frequency components of other signals except tidal signals, such as atmospheric pressure disturbance. We have added these descriptions to the revised manuscript.

6. Line 75-76, there are already many studies discussed the phase lead.

Answer: This is an error in our expression. In the original manuscript, we discussed Chinese and international situations separately. Although many international studies have discussed the phase lead regarding this question, an ideal answer to this question has not been reported in China. In the revised manuscript, we have made modifications to this statement by deleting the research situation in China.

7. Line 80-85, there are papers that modeling the tidal response with real world data. (Zhang et al., 2023; Bastias Espejo et al., 2022)

Answer: Based on the borehole lithology, the filter tubes of the casing are located at 74.20–200.07 m, with 127 mm inside diameters, where 76–84.5 m, 100–111.3 m, 118–151.25 m, and 181–200.07 m are mostly fragmented and blocky. These fragmented zones imply aquifer properties, while weak weathering and developed fractures in other sections imply semi-confined aquifer properties. The pumping experiment at the borehole completion showed that the permeability coefficient was 0.135 m/d. We consider that the model we have built is very consistent with the actual data. Furthermore, according to the analysis, the findings are also very consistent.

8. Line94, positive fracture?

Answer: We apologize for this confusion, which was based on a language translation error. We have changed "positive fracture" to "normal fault."

9. Figure 3, I would suggest to use the cpd in X-axis.

Answer: According to other suggestions, the calculation method has been changed to Fourier analysis. The unit in Figure 3 uses cpd on the X-axis.

10. Line 213, one-dimensional well-aquifer structure, could you provide a figures to show this

structure?

Answer: We have modified Figure 6 by adding a one-dimensional well-aquifer structure.

11. The conclusion section is too long and not clear, please reworded.

Answer: This is a very good suggestion. We have carefully reviewed the content of the Conclusion section again and realized that a considerable amount of the content should be moved to the Discussion section. In addition, we have reworded the Conclusion section.

12. The innovation of the study should be reworded.

Answer: We have reworded the innovation of the study; please see the end of the Introduction and Conclusion sections.

References:

Wang, C. Y., & Manga, M. (2023). Changes in Tidal and Barometric Response of Groundwater during Earthquakes—A Review with Recommendations for Better Management of Groundwater Resources. Water, 15(7), 1327.

Bastias Espejo, J. M., Rau, G. C., & Blum, P. (2022). Groundwater responses to Earth tides: Evaluation of analytical solutions using numerical simulation. Journal of Geophysical Research: Solid Earth, 127(10), e2022JB024771.

Valois, R., Rau, G. C., Vouillamoz, J. M., & Derode, B. (2022). Estimating hydraulic properties of the shallow subsurface using the groundwater response to Earth and atmospheric tides: A comparison with pumping tests. Water Resources Research, 58(5), e2021WR031666.

Gao, X., Sato, K., & Horne, R. N. (2020). General solution for tidal behavior in confined and semiconfined aquifers considering skin and wellbore storage effects. Water Resources Research, 56(6), e2020WR027195.

Lai, G., Ge, H., Xue, L., Brodsky, E. E., Huang, F., & Wang, W. (2014). Tidal response variation and recovery following the Wenchuan earthquake from water level data of multiple wells in the nearfield. Tectonophysics, 619, 115-122.

Lai, G., Jiang, C., Han, L., Sheng, S., & Ma, Y. (2016). Co-seismic water level changes in response to multiple large earthquakes at the LGH well in Sichuan, China. Tectonophysics, 679, 211-217.

Zhang, J., Liang, X., & Wang, C. Y. (2023). Capillary Impact on Tidal Response of Groundwater in Unconfined Aquifers With Finite Thickness, Anisotropy and Wellbore Storage—An Analytical Model. Water Resources Research, 59(3), e2022WR033578.

Turnadge, C., Crosbie, R. S., Barron, O., and Rau, G. C. (2019). Comparing Methods of Barometric Efficiency Characterization for Specific Storage Estimation, Groundwater, 57, 844–859, https://doi.org/10.1111/gwat.12923.

---

## Author Comment (AC4)

**水井含水层系统的潮汐响应函数**

张昭栋 郑金涵** 张广城***

（山东省地震局，济南 250021）

**摘要**  本文考虑井孔和含水层之间相互渗流的边界条件的固体潮效应、气压效应和海潮荷载效应的理论解，得出了水井含水层系统对三种不同机理的潮汐信号响应的内在联系——水井含水层系统的潮汐响应函数，其中包括水井含水层系统的幅度响应函数和位相滞后函数。本文还讨论了两种函数与水井含水层参数之间的关系及与不排水时的情况进行了比较。

**主题词：** 承压水 含水层 潮汐 响应函数 水井

**1 引言**

承压井水位对地球的固体潮、气压潮汐及海洋潮汐都有不同程度的响应。因为水井水位的潮汐幅度相当大（可达几十厘米），不用放大就可以清楚地记录到潮汐信号，因而它为观测研究固体潮提供了一种手段。另一方面，利用水位潮汐和气压效应与海潮效应，可以求出含水层的某些力学参数和渗流特性参数，所以它又为水文地质工作者提供了一种简单易行的研究含水层性质的新方法。

由于上述原因，承压井水位对潮汐的响应问题引起国内外许多科技工作者的关注，并做了不少的研究工作。但是这些工作多是建立在不排水的基础上的，所得出的井水位的固体潮系数、气压系数及海潮系数（效率）只与含水层的力学性质参数有关（如含水层的孔隙度和固体骨架的体压缩系数），而没有考虑到井孔与含水层之间的相互渗流产生的影响。

本文在前人研究的基础上，进一步考虑井孔与含水层之间的渗流作用，得出了与不排水条件下不同的固体潮系数、气压系数和海潮系数（效率），从而得出了水井含水层系统的潮汐响应函数，同时对有关问题进行了讨论。

**2 井水位对固体潮、气压和海潮的响应**

实际的水井含水层条件千差万别，本文在讨论中将其理想化，假定含水层的上下都是不透水层；各层产状水平向四周无穷延伸；含水层及其上覆地层在力学性质上都是完全弹性体；含水层本身为多孔介质，呈水饱和状态；水在含水层内的渗流是各向同性的。同时假定含水层与井孔间的滤水管对水渗流产生的阻力可以忽略不计。
* * *
* 地震科学联合基金资助课题
** 国家地震局地球物理研究所，北京 100081
*** 山东省地质矿产局

http://www.cqvip.com

由于含水层是水平无穷大薄层,所以还可以假定含水层内的水仅沿水平方向渗流,忽略垂直方向渗流的影响。

**2.1 井水位对固体潮的响应**

由文献[1—3]可知,体应变固体潮对含水层水头影响的偏微分方程是：

$$\frac{\partial H}{\partial t} = \frac{K}{S_s} \nabla^2 H - \frac{1}{S_s} \frac{\partial \Theta}{\partial t} \tag{1}$$

其中 H 为含水层内的压力水头,t 为时间,K 为含水层的渗透系数,$S_s$ 为含水层的单位贮水系数,$\Theta$ 为固体潮体应变。

对于无穷边界或有限封闭边界,并考虑到井孔与含水层之间水相互渗流的边界条件,方程(1)的解为[2,3]

$$H_w(t) = \frac{1}{S_s} A\cos \omega t + H_0 \cos(\omega t + \varphi_G + \psi) \tag{2}$$

式中

$$H_0 = \frac{r_w^2 \omega h_0 \, Ker(\alpha_k)}{2T \sin\psi} \tag{3}$$

其中 $r_w$ 为井管半径,T 为含水层的导水系数,Ker 是开尔文函数(虚宗量第二类变型贝塞尔函数的实部),

$$\alpha_k = r_w \sqrt{\frac{\omega S}{T}}$$

其中 S 为含水层的贮水系数。

由(2)式可以进一步得出单位固体潮体应变引起水井水位变化的量,即水井水位的固体潮体应变系数为

$$B_G = \sqrt{\frac{4T^2}{4T^2 - 4Tr_w^2\omega Kei(\alpha_k) + r_w^4\omega^2 K^2 ei(\alpha_k)/\cos^2\psi}} \cdot \frac{1}{S_s} \tag{4}$$

还可以导出井水位对固体潮响应的位相差为

$$\varphi_G = -arc \, tg \frac{r_w^2\omega Ker(\alpha_k)}{2T - r_w^2\omega Kei(\alpha_k)} \tag{5}$$

**2.2 井水位对气压的响应**

由文献[4]、[5]可知,气压变化对含水层水头影响的偏微分方程为

$$\nabla^2 H = \frac{S_s}{K} \frac{\partial H}{\partial t} - \frac{\alpha}{K} \frac{\partial P_a}{\partial t} \tag{6}$$

其中 $\alpha$ 为含水层固体骨架的体压缩系数,$P_a$ 为大气压力。

当考虑到井孔与含水层之间相互渗流的边界条件后,方程(6)的解为[5]：

$$H(x,y,z;t) = \frac{\alpha}{S_s} P_0 \cos \omega t + H_0 \cos(\omega t + \varphi_p + \psi) + H_a \tag{7}$$

其中 $H_a$ 为含水层内平均水头。

再进一步可以得到单位气压变化引起水井水位的变化,即水井水位的气压效率为

$$B_P = \sqrt{\frac{4T^2}{4T^2 - 4Tr_w^2\omega Kei(\alpha_k) + r_w^4\omega^2 K^2 ei(\alpha_k)/\cos^2\psi}} \cdot \frac{n\beta}{\alpha + n\beta} \tag{8}$$

其中 n 为含水层的孔隙度,$\beta$ 为含水层内水的体压缩系数。

还可以推导出井水位对气压谐谐波响应的位相差是：

$$\varphi_P = -arc \, tg \frac{r_w^2\omega Ker(\alpha_k)}{2T - r_w^2\omega Kei(\alpha_k)} \tag{9}$$

**2.3 水井水位对海潮的响应**

http://www.cqvip.com

近海的封闭含水层对海潮荷载应力也有一定的响应[6-10],可以导出海潮荷载对含水层平均应力变化对含水层水头影响的偏微分方程为[9]:

$$\nabla^2 H = \frac{S_s}{K} \frac{\partial H}{\partial t} - \frac{\alpha}{K} \frac{\partial \sigma_x}{\partial t} \tag{10}$$

考虑到井孔与含水层之间的渗流作用后,可得出方程(10)的解为[11]:

$$H_w(t) = B_m A \cos \omega t + H_0 \cos(\omega t + \varphi_s + \psi) \tag{11}$$

其中 $B_m$ 为不排水时方程(10)的解得出的水井水位的海潮荷载效率[9]。

由此可以进一步推出单位海潮荷载对水井水位的影响,即水井水位的海潮荷载效率为:

$$B_s = \sqrt{\frac{4T^2}{4T^2 - 4Tr_w^2 \omega Kei(\alpha_k) + r_w^4 \omega^2 K^2 ei(\alpha_k)/\cos^2\psi}} \cdot B_m \tag{12}$$

也可以推得水井水位对海潮荷载变化响应的位相差为:

$$\varphi_s = -arctg \frac{r_w^2 \omega Ker(\alpha_k)}{2T - r_w^2 \omega Kei(\alpha_k)} \tag{13}$$

**3  水井含水层系统的潮汐响应函数**

固体潮、气压和海潮荷载三者对承压含水层作用的机理是不同的。简单说来,固体潮对承压含水层的作用是天体起潮力使地壳含水层固体发生体积的潮汐变化,即潮汐应变。它是含水层受到一种潮汐的体积力的作用而产生的。气压潮汐变化对承压井水位的作用机理,简单地说是井孔水面和含水层上覆层地面同时受到气压潮汐变化共同作用的结果。这两个力都是面力的作用。一方面井孔水面受到气压潮汐变化的作用,例如压力增大,相当于井水面增加了一段等于气压增大量的水柱,这使得井孔内水柱压升高;另一方面地面的气压增大,这个增大的压力通过上覆层传递到含水层,使含水层内的孔隙压力升高。这个升高的孔隙压力与升高的井孔内水柱压力 要保持平衡而进行调整,调整的结果是井水位下降,这就是水井水位的气压效应。海潮荷载对承压含水层的作用与气压潮汐变化作用于地面一样,当海潮荷载增大时,传递到含水层的压力也增大,含水层内孔隙压力增大,引起水井水位升高。可见海潮荷载对承压井的作用是一个面力作用在地面的结果。

尽管以上三种现象的力的性质和作用方式不同,但三种潮汐变化都引起水井水位潮汐变化。水井含水层系统对水井水位潮汐变化的作用必定有共同之处,这就是对潮汐的幅度响应和响应位相的滞后。

比较一下固体潮系数、气压效率和海潮荷载效率三个表达式,即(4)式、(8)式和(12)式,可以找出三者的共同因式:

$$\sqrt{\frac{4T^2}{4T^2 - 4Tr_w^2 \omega Kei(\alpha_k) + r_w^4 \omega^2 K^2 ei(\alpha_k)/\cos^2\psi}} \tag{14}$$

我们把它定义为水井含水层系统的潮汐幅度响应函数,亦用 $f_s$ 表示。这样上述三个系数(效率)就可简单地写成:

$$\left. \begin{array}{l} B_G = f_s \cdot \dfrac{1}{S_s} \\[2mm] B_p = f_s \cdot \dfrac{n\beta \cdot \rho g}{S_s} \\[2mm] B_s = f_s \cdot \dfrac{\alpha \cdot \rho g}{S_s} \end{array} \right\} \tag{15}$$

http://www.cqvip.com

其中

$$S_s = \rho g(\alpha + n\beta)$$

再比较一下三种效应的位相差表达式，即(5)式、(9)式和(13)式，可以发现 3 个表达式实际上是完全一样的。可见尽管力的性质和作用方式不同，但水井水位都是潮汐变化，水井含水层对潮汐水位变化响应的位相滞后是相同的，都是由于井孔与含水层之间水渗流而引起了位相滞后。本文用 $\varphi_t$ 表示它，并称之为水井含水层系统对潮汐响应的位相滞后函数。

**4 讨论**

**4.1 水井含水层系统的潮汐幅度响应函数与水井含水层参数的关系**

水井含水层系统的潮汐幅度响应函数与水井含水层参数的关系由(14)式(即 $f_t$)决定。由于函数关系较复杂，难以直接看出 $f_t$ 与各参数的关系。(14)式中实际上有 6 个参数，它们是水井井管的半径 $r_w$、含水层的孔隙度 n、厚度 b、渗透系数 K、固体骨架的体压缩系数 $\alpha$ 以及潮汐信号的角频率 $\omega$。它们之间并非独立的，如渗透系数与孔隙度之间就有一定的关系，但它们之间的定量关系十分复杂，难以用一个代替另一个。因而给予 6 个参数可能的值，固定 6 个参数中的 4 个，即 $r_w$、K、b、n 和 $\alpha$ 中 4 个参数不变，计算出余下的一个参数与角频率之间的函数关系曲线，见图 1。图中各组曲线中的实线、虚点线和虚线分别代表该参数由小到大的三个不同数值的结果。

由图 1 可见，幅度响应函数 $f_t$ 主要取决于信号的角频率、含水层的渗透系数和厚度及井管半径。$\omega$ 愈小，K 和 b 愈大；$r_w$ 愈小，则 $f_t$ 愈大。$f_t$ 与含水层的孔隙度和固体骨架的体压缩系数关系不大。

[Figure]

**图 1 $f_t$ 与水井含水层参数关系曲线**

Fig. 1 The relationship between $f_t$ and parameters of the well aquifer.

**4.2 水井含水层系统对潮汐响应的位相滞后函数与水井含水层参数的关系**

水井含水层系统对潮汐响应的位相滞后函数 $\varphi_t$ 与水井含水层参数的关系由(5)式决定，从中难以直接看出 $\varphi_t$ 与 6 个参数的关系。与处理幅度响应函数类似，可以得到 $\varphi_t$ 与 6 个参数的关系曲线，见图 2。

由图 2 可见，$\varphi_t$ 与 6 个参数之间的关系比较复杂。$\varphi_t$ 与潮汐信号的角频率 $\omega$(或周期 $\tau$)有密切的关系。当信号周期 $\tau$ 小于 100 s 时，$r_w$ 愈小，n 愈小，则 $-\varphi_t$ 愈大；$\varphi_t$ 与 K 的关系较复杂，并且影响也较大；$\varphi_t$ 与 $\alpha$ 和 b 的关系不太大。当信号周期在 100 s 与 $10^4$ s 之间时，$r_w$ 愈大，K 愈小，b 愈小，则 $-\varphi_t$ 愈大；而 n 和 $\alpha$ 的变化对 $\varphi_t$ 的影响不大。当信号周期大于 $10^4$ s 时，b 愈小，K 愈小，$r_w$ 愈大，则 $-\varphi_t$ 愈大；而 n 和 $\alpha$ 对 $\varphi_t$ 的影响不大。

**4.3 幅度响应函数 $f_t$ 和位相滞后函数 $\varphi_t$ 与不排水解的一致性**

由文献[1]、[4]和[9]可知，在不排水的情况下固体潮系数、气压效率和海潮荷载效率分别是：

$$B_{Gn} = \frac{1}{S_s}$$

$$B_{pn} = \frac{n\beta \cdot \rho g}{S_s}$$

$$B_{sn} = \frac{\alpha \cdot \rho g}{S_s}$$

(16)

和(15)式相比较可知,两者之间只差一个系数 $f_o$。

对于长周期信号来说,角频率 $\omega \to 0$。根据三角函数和贝塞尔函数的性质,可以推得

$$\lim_{\omega \to 0} \sqrt{\frac{4T^2}{4T^2 - 4Tr_w^2 \omega Kei(\alpha_k) + r_w^4 \omega^2 K^2 ei(\alpha_k)/\cos^2\psi}} = 1$$

由此可知,当 $\omega \to 0$ 时,(15)式就变成了(16)式。也就是说,当 $\omega \to 0$ 时,即信号的变化周期变得无穷大,也就是无周期信号,此时和不排水情况相当。

当 $\omega \to 0$ 时,同样可以得到

$$\lim_{\omega \to 0}\varphi_t = \lim_{\omega \to 0} -arctg \frac{r_w^2 \omega Ker(\alpha_k)}{2T - r_w^2 \omega Kei(\alpha_k)} = 0$$

由此可见,对于长周期信号不存在位相滞后。在不排水情况下也不存在位相滞后,两者在这一方面也是一致的。

**5 结论**

在考虑到井孔与含水层之间的相互渗流影响后,本文给出了水井含水层系统对固体潮、气压和海潮荷载变化三种不同机理潮汐信号的幅度响应函数和位相滞后函数。通过本文上述对两个函数与水井含水层参数间的关系的讨论可以得出如下结论:

[Figure]

图 2  $\varphi_t$ 与水井含水层参数关系曲线

Fig. 2  The relationship between $\varphi_t$ and parameters of the well aquifer.

(1)水井含水层系统的幅度响应函数与水井含水层的参数及信号变化周期有关。当含水层的渗透系数和厚度愈大,井管半径愈小,信号的变化周期愈大,则幅度响应函数的值愈大。

(2)水井含水层系统的位相滞后函数也与水井含水层的参数及信号变化的周期有关,一般说来,井管半径愈小,含水层的渗透性能愈好,信号变化周期愈大,则位相滞后就愈小。

(3)水井含水层系统对长周期信号的响应与不排水情况一样,两者响应的幅度一样,而且都没有位相滞后。

顾功叙、陈运泰和汪成民研究员审阅了本文初稿并提出了许多宝贵意见,在此表示感谢。

http://www.cqvip.com

**参考文献**

1 张昭栋,郑金涵,冯初刚.体膨胀固体潮对水井水位观测的影响.地震研究,1986,9(4):465—472

2 张昭栋,等.深井水位的固体潮汐.地震学报,1991,13(1):66—75

3 张昭栋.地下水潮汐分析.济南:山东大学出版社,1988

4 张昭栋,郑金涵,冯初刚.气压对水井水位观测的影响.地震,1986,(1):42—46

5 张昭栋,郑金涵,张广城,靖继才.承压井水位对气压动态过程的响应.地球物理学报,1989,32(5):539—549

6 Carr P A, Kamp V D. Determining aquifer characteristics by the tidal method. Water Resources Research, 1969, 5(5):1023—1031

7 E S Robinson, R T Bell. Tides in confined well aquifer systems. J G R, 1971, 76(8):1857—1869

8 G H Rhoads, E B Robinson. Determination of aquifer parameters from well tides. J G R, 1979, 84 (B11):6071—6082

9 张昭栋,郑金涵,叶玲玲.海潮对水井水位观测的影响.地球物理学报,1990,33(专辑 11):493—500

10 叶玲玲.上海市地下水位与地表水潮汐关系的初步探讨.上海地质,1982,(4):42—49

11 张昭栋,等.承压井水位对地表潮汐的响应.地震研究,1990,13(4):377—388

**PRSPONSE FUNCTIONS OF WELL AQUIFER SYSTEM TO TIDE**

Zhang Zhaodong

(*Seismological Bureau of Shandong Province, Jinan* 250021)

Zheng Jinhan

(*Institute of Geophysics, SSB, Beijing* 100081)

Zhang Guangcheng

(*Geological and Mineral Bureau of Shandong Province, Jinan*)

**Abstract**

In this paper on the basis of the theoretical solutions of the earth tide effect, barometric effect and Oceantide load effect as the boundary conditions of permeation between a well and aquifer, the internal relations of the response of the well aquifer system to three kinds of tidal signals that have different mechanisms, i. e. the response functions of the well aquifer system to tide, have been developed, which involve the amplitude response function and phase lag function of the well aquifer system. The relationship between these two functions and the parameters of the well aquifer been discussed and compared with those under no drainage.

**Key words: Confined water, Water-bearing bed, Tide, Response function, Water well**

* Projects Sponsored by the Joint Earthquake Science Foundation